# The Impact of High-Standard Farmland Construction Policy on Rural Poverty in China

Jiquan Peng , Zihao Zhao and Lili Chen *

School of Economics, Jiangxi University of Finance and Economics, Nanchang 330013, China
* Correspondence: 822100042@stu.jxufe.edu.cn; Tel.: +86-139-8625-0230

**Abstract:** As the core component of agricultural development projects, high-standard farmland construction is a reliable measure of agricultural production, and can be used to improve the economy in rural areas. Based on provincial panel data, this paper adopts the continuous difference-in-differences (DID) method to analyze the impact of China's high-standard basic farmland construction policy on the incidence of rural poverty and its mechanisms. The results show that this policy can significantly reduce the incidence of rural poverty by 7.4%, and if, after using robust standard error and bootstrap sampling 1000 times for a robustness test, the regression results are still robust, this also shows that this inhibitory effect is stable and persistent. It can be seen from a heterogeneity analysis that the implementation of the policy has a more significant effect on poverty reduction in areas with a higher incidence of rural poverty and a larger scale of land remediation, as well as areas in the eastern and western regions. A mechanism analysis shows that natural disasters, output value and technological progress play a partial intermediary role in the poverty reduction effects of high-standard basic farmland construction policy, and the intermediary effects are 5.79%, 44.03%, and 14.13%, respectively. This paper suggests that we should continue to promote the construction of high-standard basic farmland, explore suitable construction modes of high-standard basic farmland for different regions, continuously promote the process of agricultural modernization, and broaden the ways through which rural residents are able to accumulate capital to promote rural poverty reduction and revitalization.

**Keywords:** rural poverty; agricultural investment; high-standard basic farmland; difference-in-differences method; quasi-natural experiment

## 1. Introduction

As a social phenomenon, poverty is an important problem that restricts the development of human society. The discussion on the topics of poverty and its eradication remains prominent and ongoing [1,2], especially in developing countries. As the largest developing country in the world, China's long-term dual economic structure has led to a more serious poverty problem in its rural areas [3], thus making them the main target of poverty-related governance [4]. With the implementation of a government-sponsored targeted poverty reduction strategy, China had lifted 98.99 million rural citizens out of poverty and essentially solved rural absolute poverty by the end of 2020 [5]. However, in 2021, the per capita disposable income of rural residents was 18,931 yuan, while that of urban residents was 47,412 yuan, and the income ratio of urban to rural residents was as high as 2.5 times the latter, thus indicating that rural residents are still a relatively vulnerable low-income group and that alleviating their relative poverty should be a central policy focus in the coming years.

The efficiency of farmland, as an important means of production for rural residents, determines the income and poverty of farmers to a large extent. It follows that improving the efficiency of farmland is particularly important for reducing poverty [6–8]. However, the utilization of farmland in China at present is suboptimal [9,10]. The problem of farmland fragmentation caused by the household responsibility contract system will not only

increase the cost of agricultural production, but also hinder the development of agricultural mechanization, and result in low agricultural efficiency [11,12]. Second, rapid urbanization and intensive investment in pesticides and fertilizers have led to simultaneous declines in the size and quality of cultivated land. Despite the implementation of a strict policy intended to balance the proportion of farmland, the supplementary farmland is not of sufficiently high quality, which has resulted in a downward trend in the overall quality of farmland and further kept China from reaching its full agricultural potential [13,14]. Third, infrastructure construction in rural areas of China is relatively weak and characterized by a low effective farmland irrigation rate [15], underdeveloped roads, and a water supply that is generally not centralized [16]. These problems are serious impediments to the efficiency of farmland and have thus contributed to the ongoing poverty problem in rural China.

In order to improve the efficiency of farmland use, the Chinese government has implemented an agricultural development project that promotes the construction of high-standard basic farmland (hereinafter referred to as "high-standard farmland construction") to improve overall agricultural production capacity as well as gain its social benefits. High-standard farmland, according to the General Principles for the Construction of High Standard Farmland (GB/T30600-2014), refers to farmland designated as permanently protected basic farmland for the purposes of centralization, developing new facilities, electricity matching, improving soil fertility and ecology, disaster resistance, and maximizing crop yield while generally pursuing modern agricultural production and management. With the increasingly prominent problems of agricultural ecological destruction and food security, policymakers at the national policy level have paid more and more attention to the actual progress and effectiveness of the construction of high-standard farmland. Since the beginning of the establishment of the Land Construction Development and Construction Fund, the measuring standards, construction content and task objectives of the high-standard farmland construction policy have become clearer and clearer, and the support given by the national finance has become stronger and stronger, showing obvious stage characteristics, as shown in Figure 1.

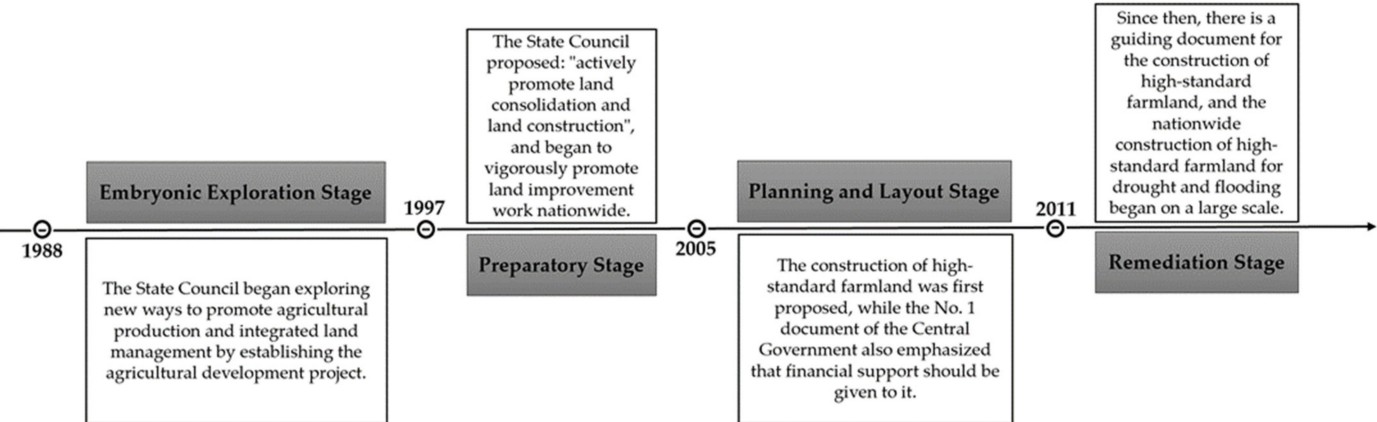

**Figure 1.** Different stages of high-standard farmland construction policy.

By 2021, 70.3 million hectares of high-standard farmland had been built in China. In the national high-standard farmland construction plan (2021–30) issued in September 2021, it is proposed that by 2025 and 2030, a total of 72 million and 80 million hectares of high-standard farmland, respectively, should be built. The construction of high-standard farmland mainly includes land leveling, disaster resistance and mitigation, and building supporting facilities, all of which can improve the profitability of agricultural production, increase crop yields, and continue to drive rural economic development [17,18]. Therefore, in the context of moving towards common prosperity, it is worthwhile to investigate whether the implementation of a the high-standard farmland construction policy will reduce poverty in rural areas and determine its mechanism of action. An exploration of the above issues will not only help to deepen our understanding of the poverty reduction

effects of high-standard farmland construction policy, but also provide a policy reference for future poverty reduction efforts in China.

The rest of this article proceeds as follows. The second section reviews the theoretical logic behind the high-standard farmland construction policy to reveal its internal mechanisms. The third section investigates the poverty reduction effects of the high-standard farmland construction policy using the difference-in-differences (DID) method. The fourth section discusses the mechanisms through which the policy reduces poverty. The fifth section summarizes the research conclusions and discusses policy recommendations.

## 2. Literature Review and Mechanisms

### 2.1. Literature Review

2.1.1. Research on Rural Poverty Reduction

Rural poverty is a complex phenomenon that is often the result of a variety of factors. The research on rural poverty is interdisciplinary and mainly focuses on three aspects. The first is economic growth and rural poverty reduction. Scholars usually regard economic growth as an important prerequisite for reducing rural poverty [19], and believe that both rural poor and non-poor groups can benefit from rural economic growth [20]. However, some scholars believe that economic growth alone does not necessarily alleviate poverty, and only "poverty-benefitting" high-quality economic growth can promote poverty reduction [21]. Furthermore, the degree of social equity is generally positively related to the poverty reduction effects of economic growth [22].

The second is income distribution and rural poverty reduction. Income inequalities will offset many of the poverty reduction effects caused by economic growth, which can at times even increase social poverty [23,24]. Ncube et al. [25] (2014) and Wan et al. [26] (2020) found that poverty reduction depends on more than the growth of average income and that income distribution plays a key role in promoting social welfare. Fang and Zhang [27] (2021) found that economic growth has contributed most to rural poverty reduction in all periods, but also that growing income inequality is detrimental to future poverty reduction. Therefore, future poverty reduction efforts should focus on pro-poor growth and income distribution equality.

The third is the relationship between public policy and rural poverty reduction. Since the reform and opening up, China's achievements in poverty reduction have mainly been the result of its rapid economic growth, regional poverty reduction and development, social security policy, and preferential policy for agriculture and land systems [28]. Therefore, many scholars have investigated the relationship between public policy and poverty reduction [29]. Xu [30] (2016) found that the agricultural income of farmers will be affected by investment efficiency of rural public goods; the higher the investment efficiency, the higher the agricultural income of farmers. Some scholars also focus on land distribution policy [31,32]. Guo et al. [33] (2019) used survey data from rural households to explain the interactions between sustainable livelihood of rural household and agricultural land transfer. Peng et al. [34,35] found that land distribution can reduce poverty by reducing both agricultural production costs and farmers' vulnerability.

2.1.2. Research on Investment in Integrated Agricultural Development and Poverty

Agricultural investment is an important measure of China's financial support for agriculture, which plays an important role in its continued development. The existing research focuses on agricultural investment and farmers' income. Studies have found that agricultural investment is not only an effective way to increase farmers' income, but also an effective tool to promote economic growth in rural areas [36,37]. However, some scholars have found that the poverty reduction effects of different sources of funding vary. For example, Guo et al. [38] (2012) found that the effects of different sources of funding in terms of increasing farmers' income are inconsistent, and the effect of fiscal funding is most significant while those of bank loans and self-raised funding are relatively weak. Yang et al. [39] (2013) found that agricultural investment has a significant impact on farmers'

incomes and confirm that the effectiveness of different sources of funding vary. There are also regional differences in the poverty reduction effects of agricultural investment. Liu et al. [40] (2014) used the DEA method to analyze the differences in fiscal expenditure in agricultural development projects by province. Their results show that the pure technical efficiency of fiscal expenditures in 18 provinces is greater than 80% and that the technical efficiency is lower in the western regions, which limits its overall efficiency. Chen [41] (2022) found that agricultural investment can significantly reduce the agricultural disaster rate and that this relationship is most prominent in the western region.

The research on farmland governance projects shows that farmland consolidation can improve rural productivity and living conditions, ensure the balance of farmland [42,43], and increase agricultural R&D investment. Together, these factors can improve agricultural production capacity and efficiency [44,45] while promoting regional agricultural economic growth. In addition, farmland governance projects can conserve labor resources by improving agricultural efficiency, thus allowing farmers to allocate surplus labor to non-agricultural production activities and achieve income growth [46,47]. The research on industrialization management shows that professional farming cooperatives and new business entities under the support of industrialization management projects can participate in processing, sales, distribution, and other agricultural processes to enhance the value of their agricultural output [48]. They can also expand the scope of industrialization, create jobs [49,50] and broaden the channels through which farmers generate income. Industrialization can also accelerate the migration of the surplus rural labor force and guide it toward non-agricultural industries to improve farmers' overall income [51,52].

In summary, the current literature on rural poverty reduction is rich. Many scholars analyze the factors that affect rural poverty reduction from the perspectives of economic growth, income distribution and public policy, but few studies focus on the relationship between agricultural investment and rural poverty. The existing research has the following shortcomings. First, it focuses on analyzing the income-increasing effects of agricultural investment and rarely discusses its poverty reduction effects. Second, it only takes agricultural investment as a whole and has not yet explored the poverty reduction effects of high-standard farmland construction. Third, most of the existing studies are theoretical analyses, while the existing empirical analyses have failed to solve the problems of model endogeneity and small sample sizes. In view of this, this paper uses provincial panel data from 2005 to 2017 to evaluate the high-standard farmland construction policy and the difference-in-differences model to analyze its impact on rural poverty and mechanisms in an effort to enrich the research on agricultural investment and rural poverty and provide valuable reference materials for high-quality agricultural development and rural poverty governance.

*2.2. Mechanism and Research Hypotheses*

2.2.1. Disaster Reduction Effects of High-Standard Farmland Construction

Due to the lack of productive assets and social capital, the ability of poorer groups to withstand the impact of extreme climate events is generally weak [53,54]. Thus, agricultural producers often suffer significant production losses [55]. Increasing extreme weather events have aggravated rural poverty. In the construction of high-standard farmland, improving farmland production capacity and farmland remediation are highly valued pursuits. In farmland construction, modern materials and equipment are customarily used to improve resistance to natural disasters and thus enhance the stability of agricultural production. At the same time, in terms of water conservancy and irrigation facilities, newly constructed high-standard farmland requires supporting facilities for field water conservancy, irrigation, and drainage projects [56]. Therefore, high-standard farmland construction can reduce the loss in agricultural production caused by uncertainties in agricultural production and management and thus improve farmers' income and reduce rural poverty.

### 2.2.2. Yield-Increasing Effects of High-Standard Farmland Construction

China is affected by the household contract responsibility system and by having an extremely high ratio of people to farmland; thus, the degree of land fragmentation is high [57]. The fragmented cultivated land structure not only limits the effective use of farmland [58] but also leads to the reduction of soil quality and affects its overall output. Moreover, fragmented land is not only not conducive to the formation of economies of scale in crop production but also inhibits technological innovation, resulting in harvest losses and increased labor costs [59]. Together, these factors exacerbate the rural poverty problem among farmers. The construction of high-standard farmland can effectively reduce the degree of fragmentation of farmland and improve farmland efficiency by incentivizing the scale effect created by continuous planting and by regulating decentralized small-scale farming operations to form concentrated and continuous high-quality farmland [60] to increase the value of agricultural output. In addition, in terms of farmland conditions and quality, the construction of high-standard farmland requires improving the thickness and depth of cultivated soil and increasing its nutrient content.

### 2.2.3. Technical Effects of High-Standard Farmland Construction

Agricultural mechanization plays an important supporting role in the transformation of agricultural production, and is the only way for China to realize agricultural modernization [61]. The use of agricultural machinery in agricultural production can significantly reduce agricultural labor costs and plant protection costs, such as weeding and pest control [62–65]. The use of agricultural machinery can also allow for multiple cropping, improve agricultural production capacity and farmland output rate [66,67] to increase the value of agricultural output [68], increase farmers' income [69], and reduce poverty. However, the use of agricultural machinery has certain requirements in terms of the farmland itself [70] and it should be noted that China's decentralized business model and low level of mechanization have greatly reduced the benefits of using agricultural machinery. One of the important goals of the construction of high-standard farmland is to transform farmland in a manner that it is suitable for mechanization, land consolidation, and the transformation of agricultural production [56] to improve operating income through technology adoption and thereby reduce rural poverty.

The theoretical framework of the high-standard farmland construction policy is shown in Figure 2. Based on the above theoretical analysis, this paper puts forward the following research hypotheses:

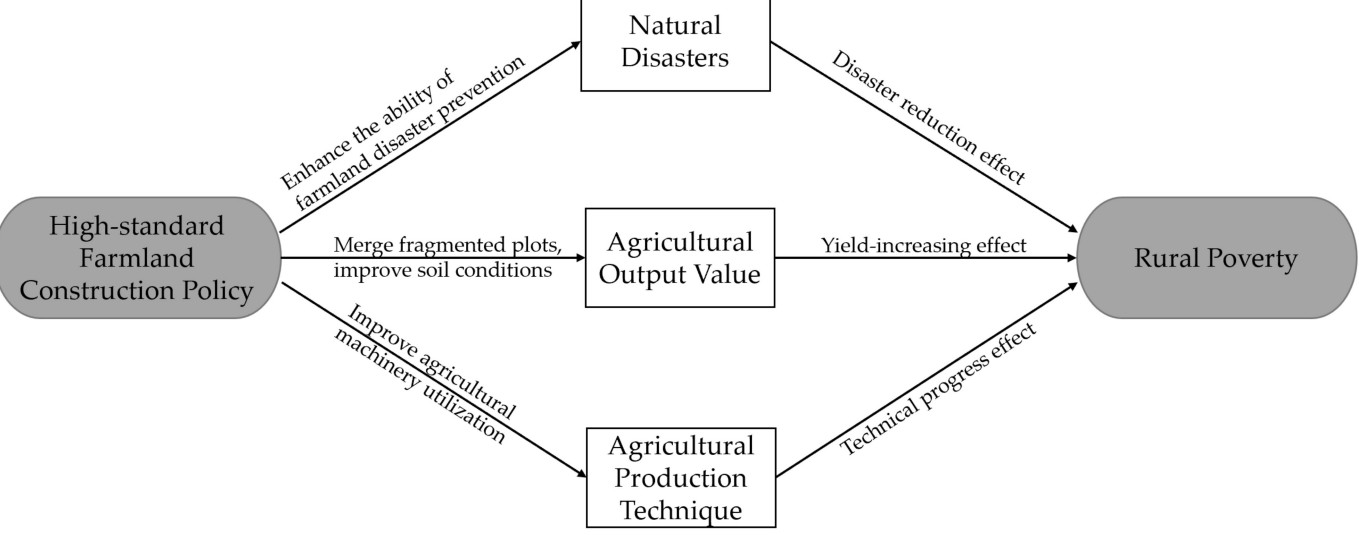

**Figure 2.** Poverty reduction mechanism of high-standard farmland construction policy.

**Hypothesis 1 (H1).** *The implementation of the high-standard farmland construction policy has curbed rural poverty.*

**Hypothesis 2 (H2).** *The high-standard farmland construction policy may reduce rural poverty by reducing agricultural disasters, increasing the value of agricultural output, and promoting technological progress.*

## 3. Research Method

### 3.1. Identification Strategy

In 2011, the high-standard farmland construction policy was implemented nationwide. This policy emphasizes that the process of high-standard farmland construction should be combined with local characteristics and be implemented gradually in different regions. Therefore, the agricultural investment of each province changes within different periods of policy implementation, and the progress is also quite different as a result. The ordinary DID model is not conducive to evaluating the poverty reduction effects of high-standard farmland construction policy. In view of this, this paper uses such policy as a quasi-natural experiment and the continuous DID model to evaluate their net effect on the incidence of rural poverty. Unlike the ordinary DID model, which uses binary dummy variables to distinguish the control and treatment groups, the continuous DID model used in this paper uses the continuous variable of agricultural investment to distinguish the control group (i.e., samples with less investment) and the treatment group (i.e., samples with larger investment). Relevant studies also show that the continuous DID model not only has the basic properties of the ordinary DID model but also can reflect changes in the degree of policy implementation, capture the heterogeneity of diverse data, and avoid errors caused by subjectively setting the control and treatment groups [71,72].

#### 3.1.1. Baseline Regression Models

In order to identify the impact of the high-standard farmland construction policy on the incidence of rural poverty, this paper constructs the following continuous DID model:

$$engel_{it} = \alpha + \beta Ai_i \times I_t^{post} + \gamma X_{it} + \delta_i + \theta_t + \varepsilon_{it} \tag{1}$$

In Equation (1), $engel_{it}$ is the explained variable that denotes the Engel's coefficient of province $i$ in period $t$, which represents its poverty level. $Ai_i \times I_t^{post}$ is the core explanatory variable, where $Ai_i$ represents the investment of agricultural development and $I_t^{post}$ represents the dummy variable at the time of policy implementation. When $t \geq 2011$, $I_t^{post}$ is 1, otherwise it is 0. $X_{it}$ is a series of control variables that includes rural medical and health levels, rural education (in years), the level of urbanization, and financial support for agriculture. The variable $\delta_i$ denotes province fixed effects and $\theta_t$ denotes year fixed effects; $\varepsilon_{it}$ is a random error term. The variable $\alpha$ is a constant term and $\beta$ and $\gamma$ are the coefficients to be estimated. This paper focuses on coefficient $\beta$ of the core explanatory variable, namely, the net effect of the high-standard farmland construction policy on the incidence of rural poverty.

#### 3.1.2. Parallel Trend Tests and Analysis of the Dynamic Effects of Policy

An important prerequisite for the use of the DID model is the satisfaction of the assumption of parallel trends; that is, the incidence of rural poverty between the control and treatment groups does not show significant differences over time before the policy was implemented. The following model is constructed to test the parallel trend hypothesis:

$$engel_{it} = \alpha + \sum_{t=2005}^{2017} \beta_t Ai_i \times D_t + \gamma X_{it} + \delta_i + \theta_t + \varepsilon_{it} \tag{2}$$

In Equation (2), $D_t$ represents the year dummy variable, and its variables and coefficients are the same as those in Equation (1). If the high-standard farmland construction policy can significantly affect the incidence of rural poverty, then before their implementa-

tion (i.e., before 2011), the coefficient $\beta_t$ should be stable; after their implementation (i.e., after 2011), the coefficient $\beta_t$ should change significantly. By using Equation (2), this paper can also estimate their dynamic impact on the incidence of rural poverty.

### 3.1.3. Mechanism Validation Model

In order to explore the internal mechanism of how the high-standard farmland construction policy affects the incidence of rural poverty, this paper further constructs the following mechanism verification model based on that proposed by Wen and Ye. [73] (2014):

In the first stage, the impact of the implementation of the high-standard farmland construction policy on the mechanism variables is verified.

$$M_{it} = \alpha + \beta Ai_i \times I_t^{post} + \gamma X_{it} + \delta_i + \theta_t + \varepsilon_{it} \tag{3}$$

In Equation (3), $M_{it}$ represents the mechanism variables, which include disaster reduction, yield increase and technological progress. The rest of the variables and coefficients are set as in Equation (1).

In the second stage, the impact of the mechanism variables on the incidence of rural poverty is verified.

$$engel_{it} = \alpha + \varphi Ai_i \times I_t^{post} + \mu M_{it} + \gamma X_{it} + \delta_i + \theta_t + \varepsilon_{it} \tag{4}$$

In Equation (4), $\varphi$ and $\mu$ are the direct and indirect effects of the implementation of the high-standard farmland construction policy on the incidence of rural poverty, respectively; the other variables and coefficients remain consistent with those in Equation (1).

## 4. Variables and Data

### 4.1. Selection of Variables

#### 4.1.1. Explained Variable

Due to the availability of data on the incidence of rural poverty in each province, the Engel coefficient is chosen as a proxy variable for the incidence of rural poverty. The choice of this proxy variable is based on the following three considerations. First, the Engel coefficient is one of the most important indicators of poverty in consumption economics, regardless of whether it is measured at the macro or micro level [74]. Second, the Engel coefficient was used by the Food and Agriculture Organization of the United Nations (FAO) in the 1970s to measure poverty in terms of national wealth and regional living standards, and it is still used in many countries and regions. For example, Lancaster et al. [75] (1999) used the Engel coefficient in a cross-country study of household budgets; Peng and Qin [76] (2021) used it as a proxy variable for the incidence of rural poverty to study the poverty reduction effects of agricultural infrastructure; Liu et al. [77] (2022) also used it to measure poverty in their study of the relationship between urbanization and poverty. In addition, the Chinese government has also included the Engel coefficient in the system of indicators for measuring the well-being of households in a moderately prosperous society [78]. Third, the Engel coefficient is a relative indicator that is not affected by purchasing power, that is, the ratio of food expenditures accounts for the total expenditure amount, and its value is not affected by monetary value.

#### 4.1.2. Core explanatory Variables

In this paper, the interaction term of the dummy variable for the year when the agricultural investment was made and the high-standard farmland construction policy was implemented ($Ai_i \times I_t^{post}$) is the core explanatory variable for assessing its impact on the incidence of rural poverty. We consider that the scale of farmland reclamation (LS) also reflects the implementation of the policy to a certain extent. Therefore, this study uses the interaction term (LS) of the sum of the change in low- and medium-yield farmland and the construction area of high-standard farmland in land governance projects as the substitution variables for the core explanatory variables; the former is logarithmically processed to eliminate heteroscedasticity.

#### 4.1.3. Control Variables

In addition to the impact of the high-standard farmland construction policy on the incidence of rural poverty, other factors also affect the incidence of poverty and therefore need to be controlled for in the model. Referring to the research of existing scholars [79,80], this paper selects the following control variables: rural medical and health levels (Medical), measured by the number of beds per capita in township health centers; rural education level (Edu), measured by average rural years of education; urbanization level (Urban), measured by the ratio of the urban population to the total population; and fiscal support for agriculture (Gov), measured by local fiscal expenditures on agriculture, forestry and water.

#### 4.2. Data Sources and Description of Characteristic Facts

#### 4.2.1. Data Sources

This paper uses panel data of 31 provinces (districts and cities) in China, excluding Hong Kong, Macao and Taiwan, from 2005 to 2017. The data come from the *China Statistical Yearbook*, *China Financial Yearbook*, *China Rural Statistical Yearbook*, and *China Civil Affairs Statistical Yearbook*. The moving average method is used to fill in the missing data. Table 1 reports the mean and variance for each variable. The average Engel coefficient is 39.08%, and the difference between the minimum and the maximum is significant. It may be that with the increase of farmers' income, the share of rural residents' food expenditures gradually narrows. The standard deviation of agricultural investment is significant, which may be related to the fiscal income gap between provinces. The standard deviation of various control variables is relatively high, and the mean value of rural education level is 8.6474 years, thus indicating that the phenomenon of low compulsory education still exists in rural areas. The standard deviation of consolidated farmland area is large, which indicates that there is a large gap in the progress of high-standard farmland construction across provinces. There are great differences in the crop disaster rate and per capita value of agricultural output among provinces, and there is little difference in the agricultural machinery per capita.

**Table 1.** Descriptive statistics.

| Variable Name | Variable Abbreviation | Metrics | Average Value | Standard Deviation |
|---|---|---|---|---|
| Incidence of rural poverty | *engel* | Food consumption expenditure/total consumption expenditure (%) | 39.08 | 0.0728 |
| Agricultural investment | *Ai* | Amount of investment in agricultural development (RMB billion) | 0.1602 | 0.0974 |
| Rural health care standards | Medical | Number of beds in township health centers/total rural population (beds/thousand people) | 1.6862 | 0.5641 |
| Rural education standards | Edu | Years of schooling for rural residents/population of school age (years/person) | 8.6474 | 1.2073 |
| Level of urbanization | Urban | Urban population/total population (%) | 0.5203 | 0.1466 |
| Level of financial support to agriculture | Gov | Local financial expenditure on agriculture, forestry and water undertakings (in billions of yuan) | 0.3164 | 0.2508 |
| Area of land reclamation | LnLS | Demonstration Project for the Improvement of Low and Medium Yield Land and High Standard Farmland (thousand hectares) | 7.1743 | 0.8807 |
| Scale of land reclamation | Hrate | Area of low- and medium-yielding land and high-standard farmland rehabilitated/total arable land area (%) | 0.3684 | 0.2373 |
| Crop failure rate | Disaster | Area affected/total crop area sown (%) | 0.1099 | 0.0860 |
| Number of agricultural machines per capita | Machine | Total number of agricultural machinery/total rural population (units/person) | 0.0346 | 0.0300 |
| Gross agricultural output per capita | LnGdp | Total value of agricultural output/total rural population (yuan/person) | 9.605 | 0.6168 |

#### 4.2.2. Descriptive Statistics

In order to preliminarily understand the dynamic relationship between policy implementation and rural poverty, this paper first conducts a statistical analysis on the relationship between core variables. Figure 3 shows that since the implementation of the

high-standard farmland construction policy in 2011, high-standard farmland construction area and agricultural investment have shown a rapid growth trend, and both have declined since 2015. The dynamic trend of the two is basically the same. Because the main role of agricultural investment is to transform low- and medium-yield fields and renovate farmland, this paper takes agricultural investment as an evaluation variable for the policy effect of high-standard farmland construction, which is a rational association. From Figure 4, it can be seen that agricultural investment showed a rapid upward trend from 2011 to 2015, while the rural Engel coefficient showed a rapid downward trend; the two exhibit an obvious inverse relationship during this period. In addition, although agricultural investment began to decline after 2015, the declining trend of the rural Engel coefficient does not change, which, to a certain extent, shows that the high-standard farmland construction policy has a persistent anti-poverty effect. The above analysis is only descriptive in nature; we discuss empirical results in a later section.

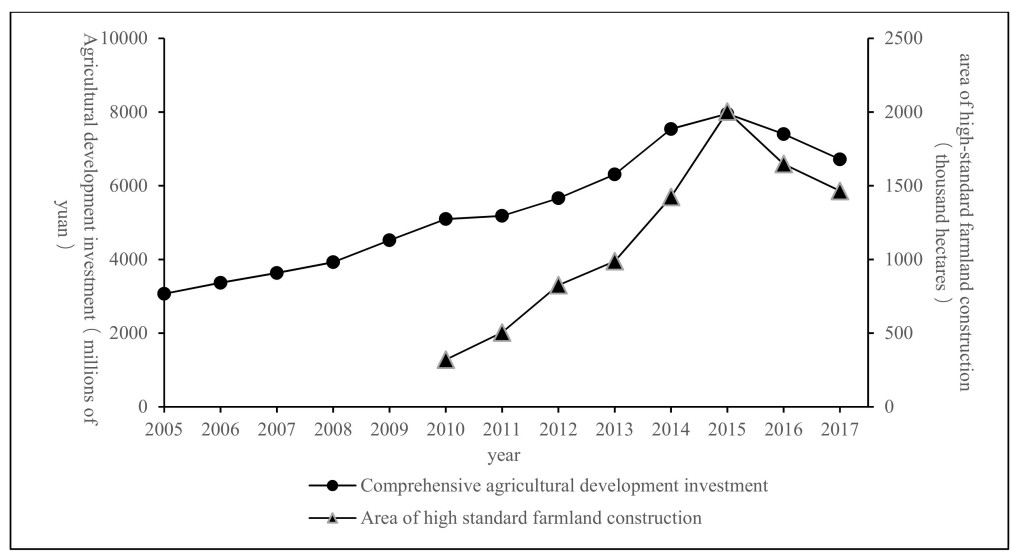

**Figure 3.** Agricultural investment and high-standard farmland construction area.

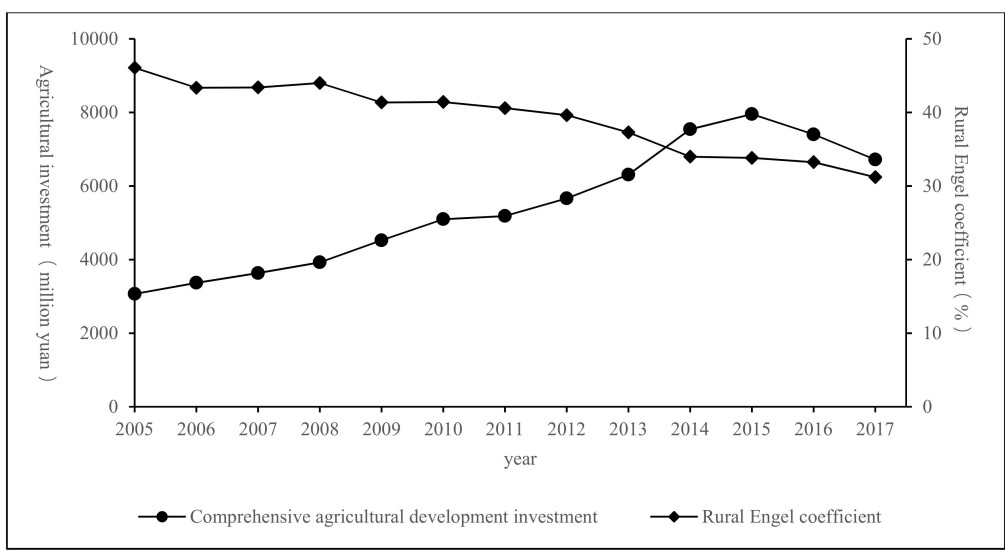

**Figure 4.** Agricultural investment and the rural Engel coefficient.

## 5. Empirical Results and Analysis

### 5.1. Baseline Regression Results

Table 2 shows the effect of agricultural investment on the incidence of rural poverty after the implementation of the high-standard farmland construction policy. In Table 2,

column (1) shows the estimation results of ordinary standard errors, column (2) shows the estimation results of the robust standard errors, and column (3) shows the estimation results of the standard errors obtained by Bootstrap self-help random sampling for 1000 iterations. The estimation results show that under the condition of simultaneously controlling the provincial and year effects, the policy's implementation has a significant indigenous impact on the incidence of rural poverty regardless of the standard errors, thus indicating that the model estimation results are relatively robust. The estimated coefficient is $-0.074$, which suggests that the implementation of the high-standard farmland construction policy has significantly reduced the incidence of rural poverty by 7.4%.

**Table 2.** Baseline estimation results.

| Engel | (1) Ordinary Standard Error | (2) Robust Standard Error | (3) Bootstrap Sampling 1000 Times |
|---|---|---|---|
| $Ai_i \times I_t^{post}$ | $-0.074$ *** | $-0.074$ *** | $-0.074$ *** |
| | (0.025) | (0.022) | (0.025) |
| Medical | $-0.015$ *** | $-0.015$ *** | $-0.015$ *** |
| | (0.004) | (0.003) | (0.004) |
| Edu | $-0.013$ ** | $-0.013$ * | $-0.013$ * |
| | (0.006) | (0.008) | (0.008) |
| lnUrban | $-0.144$ *** | $-0.144$ *** | $-0.144$ *** |
| | (0.030) | (0.039) | (0.040) |
| Gov | 0.030 * | 0.030 ** | 0.030 ** |
| | (0.000) | (0.000) | (0.000) |
| Constant | 0.456 *** | 0.456 *** | 0.456 *** |
| | (0.061) | (0.074) | (0.079) |
| Sample size | 403 | 403 | 403 |
| R-squared | 0.926 | 0.926 | 0.926 |
| Provincial effects | YES | YES | YES |
| Year effect | YES | YES | YES |

Note: ***, ** and * denote significance at the 1%, 5%, and 10% levels, respectively, and the figures in brackets are the standard errors.

With regard to the control variables, an increase of one unit in the rural medical and health level can significantly reduce the incidence of rural poverty by 0.015 units. This may be because the improvement of the medical level in rural areas can reduce the risk of poverty caused by health issues. For every unit increase in the rural education level, the incidence of poverty is significantly reduced by 0.013 units. Families with a higher education level are better at using resources for production and in operations, and thus their incidence of poverty is lower. With every one percent increase in urbanization, the incidence of poverty decreases by 0.144 percent. Scholars have found that the advancement of urbanization will promote wage growth and reduce poverty in rural households [81].

### 5.2. Parallel Trend Tests and Dynamic Policy Effects

#### 5.2.1. Parallel Trend Test

The previous section shows that the implementation of the high-standard farmland construction policy can effectively reduce the incidence of rural poverty, but the prerequisite for obtaining valid estimation results in the DID model is that it passes the parallel trend test. Therefore, this paper runs a regression based on Equation (2) to test whether the hypothesis is valid and depicts a more intuitive trend based on the regression results (see Figure 5). As can be seen from Figure 5, before the implementation of the high-standard farmland construction policy (i.e., before 2011), the regression coefficient $\beta_t$ is basically negative and shows a general downward trend. However, the regression coefficients contain zero values at the 90% confidence intervals, thus indicating that there is no significant difference between the regression coefficients before the implementation of the policy.

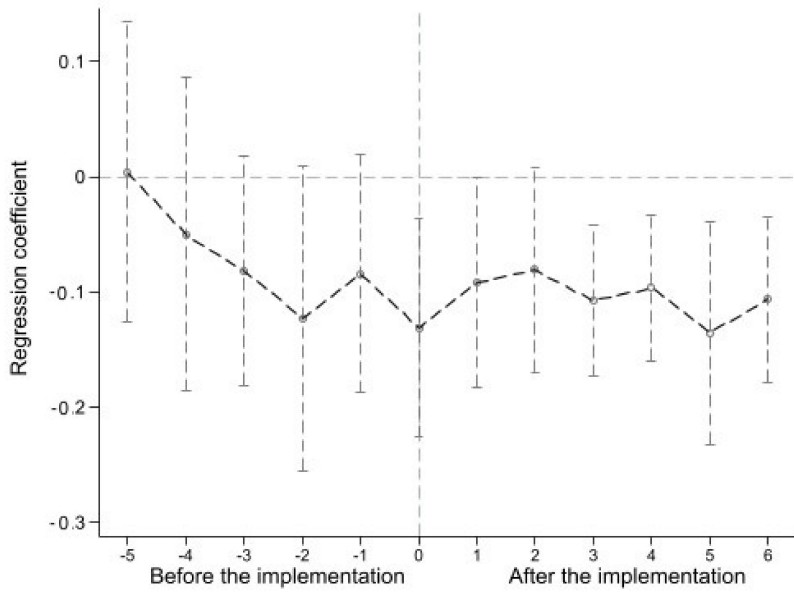

**Figure 5.** Dynamic impact of the high-standard farmland construction policy on rural poverty.

### 5.2.2. Dynamic Policy Effects

The dynamic impact of the high-standard farmland construction policy on the incidence of rural poverty is shown in Figure 5 and Table 3. Figure 5 shows that after implementation (i.e., 2011), the 90% confidence interval of the regression coefficient is below 0, which shows that it has indeed inhibited agricultural poverty. Through the regression coefficients reported in Table 3, it can be seen that the estimated coefficients for the year of policy implementation (i.e., 2011) is significantly negative at −0.131; this shows that since the implementation of policies in 2011, the high-standard farmland construction policy has played a role in poverty reduction; with the gradual expansion of the construction scale of high-standard farmland, the estimated coefficients reach their lowest value (−0.136) in 2016, which suggests that policy's poverty reduction effects are stable and sustainable.

**Table 3.** Regression results of dynamic policy effects.

| Variable | Regression Coefficient |
|---|---|
| $Ai \times 2006$ | 0.004 (0.079) |
| $Ai \times 2007$ | −0.050 (0.083) |
| $Ai \times 2008$ | −0.082 (0.060) |
| $Ai \times 2009$ | −0.123 (0.080) |
| $Ai \times 2010$ | −0.084 (0.063) |
| $Ai \times 2011$ | −0.131 ** (0.058) |
| $Ai \times 2012$ | −0.092 * (0.055) |
| $Ai \times 2013$ | −0.081 (0.054) |
| $Ai \times 2014$ | −0.108 *** (0.040) |
| $Ai \times 2015$ | −0.097 ** (0.039) |
| $Ai \times 2016$ | −0.136 ** (0.059) |
| $Ai \times 2017$ | −0.106 ** (0.043) |
| Constant | 0.467 *** (0.075) |
| Sample size | 403 |
| R-squared | 0.927 |
| Control variables | YES |
| Provincial effects | YES |
| Year effect | YES |

Note: ① ***, **, * denote significance at the 1%, 5%, and 10% levels, respectively; ② figures in brackets are the standard errors.

### 5.3. Robustness Tests

The above results show that the high-standard farmland construction policy effectively reduces the incidence of rural poverty and has a lasting policy effect. However, the results may still be confounded by omitted variables and sample self-selection. In order to improve the robustness of the model estimation, this section draws on existing research [82,83], and conducts robustness tests by replacing the core explanatory variables, changing the sample period and one-period lagging the control variables. The test results are shown in Table 4.

**Table 4.** Robustness test results.

| Variable | (1) Replacement of Core Explanatory Variables | (2) Change of Sample Period | (3) Control Variables Lagged by One Period |
|---|---|---|---|
| $\ln LS \times I_t^{post}$ | −0.0137 *** | | |
| | (0.004) | | |
| $Ai_i \times I_t^{post}$ | | −0.0661 ** | −0.0711 *** |
| | | (0.031) | (0.023) |
| Control variables | YES | YES | |
| Control variable lag | | | YES |
| Constant | 0.4568 *** | 0.4518 *** | 0.4708 *** |
| | (0.073) | (0.094) | (0.081) |
| Sample size | 403 | 217 | 372 |
| R-squared | 0.9276 | 0.9148 | 0.9265 |
| Provincial effects | YES | YES | YES |
| Year effect | YES | YES | YES |

Note: ① *** and ** denote significance at the 1% and 5% levels, respectively; ② figures in brackets are the standard errors.

(1)  Substitution of core explanatory variables: We consider that the extent of high-standard farmland construction can be characterized not only by using agricultural investment but also by using the area of farmland remediation (LS). Therefore, the interaction term of the dummy variable of land remediation area and the year of policy implementation was selected as a proxy for the core explanatory variables. The results are shown in column (1) of Table 4, where the regression coefficient of the new interaction term is −0.0137 and is significant at the 1% level, thus indicating that policy implementation still has a significant effect on rural poverty.

(2)  Changing the sample period: Because the previous regression results were based on the full sample, the high-standard farmland construction policy went into effect in 2011, and thus there is a longer period before its implementation. In order to ensure that there is little difference in the periods before and after implementation, a sample from 2009–15 (i.e., two years before and four years after implementation) is selected for further analysis, which can also partly avoid the impact of the 2008 financial crisis. The results are shown in column (2) of Table 4. The regression coefficient is −0.0661, and the estimated results are consistent with those generated by the dominance test.

(3)  Lagging the control variables by one period: Considering the possible causal relationship between the high-standard farmland construction policy and the incidence of rural poverty, all control variables were regressed with a one-period lag in order to weaken the potential endogeneity effect. The results are shown in column (3) of Table 4. The sign and significance of the regression coefficients remain consistent with the previous baseline regression results, again verifying the robustness of the baseline regression results.

### 5.4. Heterogeneity Analysis

(1)  Heterogeneity of different poverty levels: Considering the difference in the incidence of poverty across regions, the role of the high-standard farmland construction policy may also be different. In order to test the heterogeneity of policy implementation

in regions with different poverty levels, the samples are divided into low poverty incidence and high poverty incidence groups according to the intermediate quantile of poverty incidence, and then further analyzed. As shown in columns (1) and (2) of Table 5, the policy effect in regions with high poverty incidence is significantly greater than that in regions with low poverty incidence. One possible explanation for this result is that the principle of "giving priority to the old revolutionary and poor areas" was put forward in the "Opinion on effectively strengthening the construction of high-standard farmland to enhance the national food security guarantee capacity[1], so the construction of high-standard farmland in areas with high incidence of poverty may be faster than that in areas with low incidence of poverty, and its effect on poverty reduction more significant.

(2)   The heterogeneity of land consolidation: The scale of land consolidation in various regions reflects the progress of high-standard farmland construction to a certain extent, and different levels of construction progress will also have different poverty reduction effects. In order to test the heterogeneity of the policy effects under different land consolidation scales, the samples are divided into large and small farmland consolidation scale groups according to the middle quantile of farmland consolidation. From column (3) and column (4) in Table 5, it can be seen that although the regression coefficient of the small-scale farmland consolidation group is negative, it does not pass the dominance test. The regression coefficient of the large-scale farmland consolidation group is significantly negative, which indicates that with the deepening of high-standard farmland construction, the poverty reduction effects of the policy become stronger.

(3)   The heterogeneity of different geographic locations: Taking into account the differences in climatic conditions, soil quality, and economic development across regions, the samples are divided into eastern, central and western regions according to the classification criteria of China National Development and Reform Commission. Table 6 reports the impact of the high-standard farmland construction policy on the incidence of poverty in the three regions. The results show that in the eastern and western regions, the impact coefficient is significantly negative, and the regression coefficient in the western region is smaller than that in the eastern region, thus indicating that the policy reduction effect in the western region is stronger. The estimation results of the central region are not apparent, which may be due to the existence of its large agricultural surplus labor force. These labor resources cannot be transferred in the short term, which is conducive to poverty and offsets the effect of the policy. The agricultural development conditions in the western region are worse than those in the eastern region. The implementation of the high-standard farmland construction policy can therefore have the most prominent poverty reduction effects in the western region.

**Table 5.** Results at different poverty levels and land consolidation scales.

| | (1) Lower Poverty Incidence Group | (2) Higher Incidence of Poverty Group | (3) Small-Scale Land Consolidation Group | (4) Large-Scale Land Consolidation Group |
|---|---|---|---|---|
| $Ai_i \times I_t^{post}$ | −0.0396 ** (0.018) | −0.1797 ** (0.080) | −0.0348 (0.031) | −0.1051 *** (0.036) |
| Constant | 0.3007 *** (0.071) | 0.6315 *** (0.121) | 0.2387 ** (0.108) | 0.3495 *** (0.075) |
| Sample size | 202 | 202 | 202 | 202 |
| R-squared | 0.8891 | 0.8523 | 0.9379 | 0.9564 |
| Control variables | YES | YES | YES | YES |
| Provincial effects | YES | YES | YES | YES |
| Year effect | YES | YES | YES | YES |

Note: ① *** and ** denote significance at the 1% and 5% levels, respectively; ② figures in brackets are the standard errors.

**Table 6.** Results in different natural geographical locations.

| | (1) Eastern Region | (2) Central Region | (3) Western Region |
|---|---|---|---|
| $Ai_i \times I_t^{post}$ | −0.1155 *** (0.036) | 0.0257 (0.034) | −0.1575 *** (0.060) |
| Constant | 0.4418 *** (0.084) | 0.3813 ** (0.148) | 0.4693 *** (0.139) |
| Sample size | 143 | 104 | 156 |
| R-squared | 0.9537 | 0.9313 | 0.9159 |
| Control variables | YES | YES | YES |
| Provincial effects | YES | YES | YES |
| Year effect | YES | YES | YES |

Note: ① *** and ** denote significance at the 1% and 5% levels, respectively; ② figures in brackets are the standard errors; ③ The eastern region is composed of 11 provinces and cities, namely Beijing, Tianjin, Hebei, Liaoning, Shanghai, Jiangsu, Zhejiang, Fujian, Shandong, Guangdong and Hainan; the central region includes eight provinces and cities in Shanxi, Jilin, Heilongjiang, Anhui, Jiangxi, Henan, Hubei and Hunan; the western region includes 12 provinces and cities in Inner Mongolia, Guangxi, Chongqing, Sichuan, Guizhou, Yunnan, Tibet, Shaanxi, Gansu, Qinghai, Ningxia and Xinjiang.

*5.5. Further Analysis: Mechanism Analysis*

The empirical results show that the high-standard farmland construction policy can significantly reduce the incidence of rural poverty. However, the pathway through which the policy takes effect remains unclear. In order to solve this problem, this paper further explores its mechanisms. The National High Standard Farmland Construction Plan (2021–30) issued by the Ministry of Agriculture and Rural Affairs of China shows that high-standard farmland construction is conducive to improving the agricultural production environment, increasing agricultural output, and promoting the transformation of agricultural production methods. Therefore, this paper assumes that the poverty reduction mechanism of high-standard farmland construction may be that policy implementation can reduce agricultural disasters, increase the value of agricultural output, and promote progress in agricultural production. In order to verify the above conjecture, a regression analysis is carried out based on Equations (3) and (4). Table 7 shows the estimation results.

**Table 7.** Regression results for the mechanism analysis.

| | (1) Disaster | (2) lnGdp | (3) lnMachine | (4) | (5) Engel | (6) |
|---|---|---|---|---|---|---|
| $Ai_i \times I_t^{post}$ | −0.127 * (0.067) | 0.713 *** (0.200) | 1.002 *** (0.235) | −0.0695 *** (0.022) | −0.0413 * (0.022) | −0.0633 *** (0.023) |
| Disaster | | | | 0.0336 ** (0.017) | | |
| lnGdp | | | | | −0.0455 *** (0.011) | |
| lnMachine | | | | | | −0.0104 ** (0.005) |
| Constant | 0.0488 (0.29) | 9.1165 *** (0.498) | −1.843 ** (0.716) | 0.454 *** (0.074) | 0.871 *** (0.139) | 0.437 *** (0.077) |
| Sample size | 403 | 403 | 403 | 403 | 403 | 403 |
| R-squared | 0.5060 | 0.9608 | 0.9292 | 0.9809 | 0.9315 | 0.9498 |
| Control variables | YES | YES | YES | YES | YES | YES |
| Provincial effects | YES | YES | YES | YES | YES | YES |
| Year effect | YES | YES | YES | YES | YES | YES |

Note: ① ***, **, * denote significance at the 1%, 5%, and 10% levels, respectively; ② figures in brackets are the standard errors.

(1) Disaster-mitigating effects of high-standard farmland construction policy:

The crop disaster rate was selected to measure agricultural disasters, and columns (1) and (4) in Table 7 show the estimation results of the disaster reduction effect. From the estimation results in column (1), it can be seen that the regression results on the crop

disaster rate are significantly indigenous and the coefficient is negative, thus indicating that the high-standard farmland construction policy has a significantly negative impact on agricultural disasters. Column (4) shows that the crop disaster rate has a significantly positive impact on rural poverty, and the absolute value of the estimated coefficient on the policy variable decreases from 0.074 in column (1) of Table 2 to 0.0695 in column (4) of Table 7, which suggests that the crop disaster rate has a partial mediating effect in the policy impact on the incidence of rural poverty. Its mediating effect is 5.79%($-0.127 \times 0.0336/-0.074$).

(2)  Yield-enhancing effects of the high-standard farmland construction policy:

The above has confirmed the disaster reduction mechanism of the high-standard farmland construction policy and examines its impact on agricultural production. The total value of agricultural output is selected as an indicator to measure agricultural production. Columns (2) and (5) in Table 7 show that the policy has a significantly positive impact on the total value of agricultural output, thus indicating that high-standard farmland can significantly improve farmland productivity. In addition, by incorporating policy variables and gross agricultural output into the model, it can be seen that the effect of increased agricultural production on the incidence of rural poverty is partially mediated by the policy of high-standard farmland construction, with a mediating effect size of 44.03%($0.713 \times -0.0455/-0.074$). Therefore, we confirm that the high-standard farmland construction policy can reduce the incidence of rural poverty by increasing agricultural output.

(3)  Technological advancement effect of the high-standard farmland construction policy:

Agricultural mechanization can improve the operating conditions and technological level of agricultural production [84,85]. In this paper, the number of agricultural machines per capita is used as an alternative variable for the level of agricultural mechanization to measure agricultural technological progress. Columns (3) and (6) in Table 7 show that the high-standard farmland construction policy has a significantly positive impact on the number of agricultural machines per capita, thus indicating that it can effectively promote agricultural technological progress. Furthermore, the coefficient on agricultural machines per capita is significantly negative, which suggests that agricultural technological progress has a partial mediating effect in the influence of the high-standard farmland construction policy on the incidence of rural poverty. It can be seen that technological progress partially mediates the effect of high standard farmland construction policy on rural poverty incidence, with a mediating effect size of 14.13%($1.002 \times -0.0104/-0.074$). Thus, the poverty reduction mechanisms of the high-standard farmland policy have been identified, namely, that the implementation of the policy will reduce agricultural disasters, increase agricultural output and promote agricultural progress, thereby reducing rural poverty.

## 6. Conclusions and Recommendations

Poverty is a chronic disease in human society, and poverty's governance is inextricably linked to happiness. This paper takes the high-standard farmland construction policy implemented in China in 2011 as a quasi-natural experiment. Based on China's provincial-level panel data from 2005 to 2017, the continuous DID model is used to estimate the poverty reduction effects of the implementation of the high-standard farmland construction policy in rural areas. The results show the following findings. First, the policy can reduce the incidence of rural poverty by 7.4%. After a series of robustness tests, we find that the poverty reduction effects of the policy hold. Second, the policy's effects in regions with a high incidence of poverty is significantly greater than that in regions with a low incidence of poverty These effects are not immediate and are only observed when construction reaches a certain scale; they are more obvious in the eastern and western regions, and are not as obvious in the central region. Third, the incidence of poverty in rural areas can be reduced by reducing agricultural disasters, increasing agricultural output and promoting progress in agricultural production.

These conclusions show that the implementation of the high-standard farmland construction policy has reduced poverty in rural areas, which not only supports rural revi-

talization but also promotes common prosperity. Therefore, it is necessary to give full play to the poverty reduction effect of high-standard farmland construction and accelerate its implementation. Specifically, first, it is necessary to adhere to the implementation of the policy, strengthen the organization and implementation of high-standard farmland construction projects, and seize the farming season, good timing, and key nodes to speed up the construction progress of the project. It is also necessary to actively explore diversified financing mechanisms such as financial subsidies, investment subsidies, and social capital, make good use of government bonds and farmland transfer income, and give priority to the adjusted income generated by new farmland in high-standard farmland construction, thus making greater use of the high-standard farmland construction policy's poverty reduction role. Second, there is a need to increase the construction of high-standard farmland in areas with low poverty incidence and in central regions in order to change the situation that the policy has little effect in these areas, so as to expand the positive impact of the policy on rural poverty reduction and rural revitalization. In the meantime, the policy should be flexibly adjusted to explore its suitability for different regions. For the eastern region, with its high level of agricultural mechanization and specialized production, the research and promotion of new agricultural science and technology should be strengthened. For the central region, with its heavy rainfall and sandy soil, it is necessary to vigorously promote water-saving irrigation technology, improve water utilization efficiency, and adopt water-saving irrigation measures, such as channel seepage prevention and pipeline water conveyance irrigation, according to local conditions. For the western region, which is dominated by sloping farmland, attention should be paid to the construction of local agricultural infrastructure and the promotion of practical agricultural technologies, such as high-yield cultivation techniques and agricultural biotechnology. Third, we recommend that improving the ability of disaster prevention, promoting agricultural income and production, and modernizing agricultural machinery should be the focus of future high-standard farmland construction policy. In the construction of high-standard farmland, attention should be paid to the construction of supporting facilities such as hydropower, field roads, and irrigation ditches to enhance the ability of farmland to resist disasters; at the same time, relevant departments must vigorously promote agricultural social services, such as irrigation and drainage, fertilizer distribution and management, pest control and other services, to achieve the goal of agricultural production efficiency, and improve agricultural management income. It is also necessary to strengthen the application of agricultural science and technology and promote good varieties and methods. At the same time, we should promote a combination of engineering, agronomy and agricultural machinery technologies, strengthen the technical environment, realize the "three networks" of fields, canals, and roads, promote mechanization, scale and standardization, and improve the "three forces" of farmland drainage and irrigation capacity, farmland production capacity, and agricultural machinery operational capacity.

At present, a large number of scholars have summarized China's rural poverty reduction experience, from the macro level, such as in context of the economic system, poverty alleviation policies, income distribution, and social security [21,26,30]; to the micro level, such as cultivation, structure, agricultural technology progress, human capital and farmland transfer [31,46,60,68]. However, the attribute of farmland as an important material capital for farmers to increase their income and reduce poverty has remained unchanged [6–8]; just as Andersen [37] (2015), Bayes [46] (2001), and Renkow et al. [47] (2003) have found that farmland reclamation projects can improve agricultural productivity and contribute to poverty reduction, the study in this paper confirms these conclusions. However, the above scholars have failed to focus on the poverty reduction effect of such farmland remediation projects as the construction of high-standard farmland, which is currently the largest, highest specification and most invested land consolidation project in China, and its research significance is greater. In addition, scholars in the past mainly thought that poverty reduction through farmland consolidation was achieved by increasing agricultural

production [44,45], which is also confirmed in this paper, but farmland consolidation should promote poverty reduction from multiple dimensions, which is also a focus of this paper.

This study has certain limitations. The first is the timeliness of the data. Since the National Bureau of Statistics does not publish data on investment in integrated agricultural development in the provinces after 2017, the data used in this paper span from 2005 to 2017 and may not be able to assess recent policy effects. The second is regional heterogeneity. According to the National High Standard Farmland Construction Plan (2021–30), the construction area is mainly divided into seven regions: Northeast region, Huang-Huai-Hai region, Middle and lower reaches of the Yangtze River, Southeast region, Southwest region, Northwest region and Qinghai-Tibet region. However, due to the division of the northeast region, it does not include the whole region of Inner Mongolia and thus it is impossible to use the existing provincial-level data in this research. The method for dividing regions into the east, middle and western regions may not be detailed enough in our analysis of regional heterogeneity. This paper's unique contribution is its discussion of the internal mechanisms of poverty reduction to deepen our understanding of the practical significance of agricultural investment. These problems can be further studied in future research efforts.

**Author Contributions:** Conceptualization, J.P. and L.C.; data curation, Z.Z.; formal analysis, J.P. and Z.Z.; investigation, J.P.; methodology, Z.Z.; project administration, J.P.; writing—original draft, Z.Z.; writing—review and editing, L.C. All authors have read and agreed to the published version of the manuscript.

**Funding:** National Natural Science Foundation of China (No. 72063012); MOE (Ministry of Education in China) Humanities and Social Sciences Youth Foundation (No. 20YJC790103); General Project of Science and Technology Research of Jiangxi Provincial Department of Education (No. GJJ210526); Jiangxi Province Postdoctoral Research Project (No. 2019KY33).

**Data Availability Statement:** All data and materials are available upon request.

**Acknowledgments:** We would like to thank the reviewers for their thoughtful comments that helped improve the quality of this work.

**Conflicts of Interest:** The authors declare no conflict of interest.

## Notes

[1]    Document: 'Opinion on effectively strengthening the construction of high-standard farmland to enhance the national food security guarantee capacity' National Office [2019] No. 50.

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
