# Peer review of "The Impact of High-Standard Farmland Construction Policy on Rural Poverty in China"

_land, doi:10.3390/land11091578_

Round 1
Reviewer 1 Report
The research is very interesting and, after some modifications, it may be published in the journal. The following items require corrections.
1. The abstract should reflect the statistical nature of the study.
2. In order to understand the entire context of the research, please present the stages of the conducted research or a respective flowchart. In order to systematize the Research Method, schould be described in one chapter. Does the description of the method also include the results? The text needs to be reorganized.
3. The summary and conclusions are very clearly presented – no comments.
Reviewer 2 Report
The paper has talked about a very important issues in China, and the author has also used some reasonable empirical methods to analysis the problem and find some useful results and the corresponding policy implication, while some major problem concerns from my point.
(1) The first problems lies in: the author haven’t put forward some meaningful policy implications for the current land policy in rural China, as the author said, we should continue to promote the construction of high-standard basic farmland, this is none sense, the author has talked about three intermediary effects by three important factors, while has not put forward some issues regarding on these, but talked about “to accumulate capital of the rural residents”, that is to say, your policy recommendation is not from what your results find.
(2) Generally speaking, as a scientific research article, after you has arisen a good research question, you should try to construct a “theoretical” or “conceptual” framework, in your paper, you haven’t tried to construct this conceptual framework, this theoretical framework should discussed the impact mechanism of the high-standard basic farmland on rural poverty, a diagram to show the influence mechanism, routes and “effects”, which helps you to construct some research hypothesis, unfortunately, I haven’t seen this parts.
(3) Part 2.3 is the most important in this paper, the author has construct three effect which called “disaster reduction effect”, “Yield-increasing effects” and “technical effects’, as I Can see, it is mostly comes form the literature summarization, not comes from some valid theories, I suggest the author construct a conceptual framework, which use a “diagram”, use some figure to express the influence mechanism, in addition, due to your contribution is reveal the internal influence mechanism, I consider that you put forward three hypotheses, which discuss the influence routes and directions of the three effects respectively.
(4) In methodology section, the author has used some good econometric empirical method to tested and verify the research hypotheses. However, when talk about the rural poverty, it is a big “concept”, the author seems has not found the most reasonable variable to measure this indicator, as I can concerned, the Engel coefficient normally respect the percentage of farmers who spend the money on food occupy the expenditure on living, which we used to measure one’s living standards both in China and western literature, it is not very reasonable in measure rural poverty, the best indicators may should be rural residents’ income (different income types).
(5) In fact, as the author’s empirical test, the policy establishment effect has gone into effect in 2011, that is to say the high-standard farmland construction policy has its response stage, the author should say and discuss some parts regarding on this, try to explain the possible reason that why it needs 4-5 years to visible the policy effect.
(6) In the robust check section, the author has discussed three methods, while lagging the control variables by one period, the author needs to give some explanation, to discuss why it is valid and convincing.
(7) The context construction seems to be confused, in section 5, it should not be Mechanism analysis, it should be result analysis, in this part you can talk about the results, whether it has verified the influence mechanism that you have talked in your theoretical parts, and what can you learn from the regression results. In addition, the sections 5 mechanism analysis should go into the section 4, which you should discussed very clearly in empirical results and analysis.
(8) The author should also give some explanations about the significant partial mediating effect of 44.03% and mediating effect of 14.13%.
(9) The author has discussed the three intermediate effects, while in your conclusion and policy recommendation, you should mostly focus on the three effects, talked about disaster reduction, yield increasing and agricultural technology adoption, and extend the policy effects which can be enjoyed by the rural residents.
(10) The author has used data, empirical methods, and literature supporting to responded a current practice problem, while the theoretical analysis is sufficient, and also the context should be reduced, the structure should be adjusted, the literature review should be more concise.
Round 2
Reviewer 2 Report
The Author has done a lot of modifications for the manuscript and also give some response to my recommendations and suggestions: while, it seems still not very sufficient:
1 In point 8: I wrote, The author should also give some explanations about the significant partial mediating effect of 44.03% and mediating effect of 14.13%. I think you should give very specific explanation, with the regression results in the above table and to explain, use the coefficient and tell the reader how do you calculate the total effect? What is the mediating effect? And what is the direct effect? You must explain it very clearly.
2 you still use mechanism analysis in section 5.5, As I have told you, it shouldn’t be mechanism analysis, it can be result analysis. Normally the theoretical mechanism should put in section 2, the theoretical mechanism and hypotheses section, you discuss the mechanism and then you use the data to verify your mechanism.
3 In sections 2.3, you write methods……, actually, you should not put methods in this part, it should put in section 3 , the empirical research section.
4 the policy review in section 2 should put in section 1 in the introduction, it is the introduction of the policy process, and then you can put forward your focused research question.
5, you should tell us, the high-standard farmland construction policy has its response stage, the author should say and discuss some parts regarding on this, try to explain the possible reason that why it needs 4-5 years to visible the policy effect.
What I want to see, in the regression results, that the regression coefficients before 2016 are not significant or at a low level and then after 2016 the coefficients are big and significant after 2016.
The revision should be sincerely, pay attention and detail and must be the best responsibility.
